# Auditory Steady-State Responses in Schizophrenia: An Updated Meta-Analysis

**DOI:** 10.3390/brainsci13121722

**Published:** 2023-12-16

**Authors:** Inès Zouaoui, Alexandre Dumais, Marc E. Lavoie, Stéphane Potvin

**Affiliations:** 1Centre de Recherche de l’Institut Universitaire en Santé Mentale de Montréal, Montreal, QC H1N 3V2, Canada; ines.zouaoui@umontreal.ca (I.Z.); alexandre.dumais@umontreal.ca (A.D.); marc.lavoie@teluq.ca (M.E.L.); 2Department of Psychiatry and Addiction, Faculty of Medicine, University of Montreal, Montreal, QC H3T 1J4, Canada; 3Institut National de Psychiatrie Légale Philippe-Pinel, Montreal, QC H1C 1H1, Canada; 4Département de Sciences Humaines, Lettres et Communication, Université TÉLUQ, Montreal, QC G1K 9H6, Canada

**Keywords:** schizophrenia, ASSR, electroencephalography, magnetoencephalography, gamma rhythm, meta-analysis

## Abstract

This meta-analysis investigates auditory steady-state responses (ASSRs) as potential biomarkers of schizophrenia, focusing on previously unexplored clinical populations, frequencies, and variables. We examined 37 studies, encompassing a diverse cohort of 1788 patients with schizophrenia, including 208 patients with first-episode psychosis, 281 at-risk individuals, and 1603 healthy controls. The results indicate moderate reductions in 40 Hz ASSRs in schizophrenia patients, with significantly greater reductions in first-episode psychosis patients and minimal changes in at-risk individuals. These results call into question the expected progression of ASSR alterations across all stages of schizophrenia. The analysis also revealed the sensitivity of ASSR alterations at 40 Hz to various factors, including stimulus type, level of analysis, and attentional focus. In conclusion, our research highlights ASSRs, particularly at 40 Hz, as potential biomarkers of schizophrenia, revealing varied implications across different stages of the disorder. This study enriches our understanding of ASSRs in schizophrenia, highlighting their potential diagnostic and therapeutic relevance, particularly in the early stages of the disease.

## 1. Introduction

Auditory hallucinations are prevalent symptoms of schizophrenia, reported in 60 to 80% of patients [1]. These hallucinations are hypothesized to be closely related to a neural imbalance between excitation and inhibition, particularly involving synaptic interactions between parvalbumin interneurons and pyramidal neurons [2]. Emerging evidence suggests that this excitation–inhibition imbalance is not merely the result of GABAergic dysfunction but also involves deficits in glutamatergic NMDA receptors on parvalbumin-positive GABAergic interneurons, leading to alterations in gamma oscillations [3,4]. Moreover, these dysfunctional gamma-band oscillations, specifically in the 30–100 Hz range, have been linked to cognitive deficits such as impaired information processing speed and working memory, commonly observed in schizophrenia [5,6]. Therefore, the role of gamma-band oscillations becomes a central focus in understanding the pathophysiology of schizophrenia.

Auditory Steady-State Responses (ASSRs) are electrophysiological markers of auditory system activity, often observed in the gamma frequency range (30–100 Hz) with a robust response at 40 Hz [7]. Generated in response to periodic auditory stimuli, such as click trains or amplitude-modulated tones, ASSRs primarily consists of two components: power and phase. Power provides insights into the intensity of neural oscillatory activity, while phase gives information about the timing of these oscillations relative to the auditory stimulus. These components can be measured using non-invasive electrophysiological tools such as electroencephalography (EEG) and magnetoencephalography (MEG) [8]. Neural sources of ASSRs have been mainly located in brain regions such as the medial primary auditory cortex, the superior temporal gyrus, and Heschl’s gyrus—areas frequently implicated in auditory processing [9,10,11,12,13].

The 40 Hz ASSR, with its capacity to quantitatively measure gamma-band oscillations, has been increasingly acknowledged as a possible biomarker for evaluating the underlying neural mechanisms of excitation/inhibition imbalance in schizophrenia [14,15]. This result has been supported by a meta-analysis by Thuné et al. [16], which synthesized outcomes from 20 EEG and/or MEG studies, demonstrating reduced power and phase in response to 40 Hz ASSRs among schizophrenia patients relative to healthy controls.

Numerous new studies have been published since that first meta-analysis [16]. This surge in available data offers an opportunity for enhanced analysis. While their work made valuable contributions by highlighting the reduced power and phase of 40 Hz ASSRs in schizophrenia, it also included some exploratory research on beta oscillations at 20 Hz. However, a more in-depth study on 20 Hz and any investigation of 30 Hz were notably absent, likely due to insufficient data focusing on these frequencies. The investigation of 20 Hz is of methodological interest, as it may allow us to determine the specificity of gamma oscillations (30–100 Hz range) compared to beta oscillations (20 Hz). Additionally, the available data offer an opportunity to look at specific subgroups, such as those with first-episode psychosis (FEP) and at-risk populations. Recent studies have shown deficits in ASSRs in these populations [14]. It remains to be determined, however, whether the amplitude of ASSR alterations exhibit a gradient effect (SCZ > FEP > at risk), similar to what has been observed in the case of other potential biomarkers of schizophrenia [17,18].

Expanding on the previous meta-analysis, we aim to investigate the same variables further: stimulus duration, stimulus type (click vs. amplitude tones), and analysis level (sensor vs. source). The distinction between sensor and source and between EEG and MEG is important. Edgar et al. [19] demonstrated that sensor-based evaluations have limitations, like the inability to discern hemisphere variations in 40 Hz responses, whereas source-based assessments in regions like the STG provide more precise and region-specific insights. In addition, we planned to look at variables that have not been fully explored before, including the effects of symptom severity, medication dosage, and attention load during stimuli presentation. Understanding the role of these factors is crucial. If the ASSRs are reduced in SZ, regardless of where the person’s attention is, it suggests that ASSRs might indicate a basic sensory deficit in psychosis rather than a problem with adjusting neuromodulation based on the context [20,21].

Therefore, our study aims to offer a comprehensive and updated meta-analysis that synthesizes these new data, enhancing our understanding of the utility of ASSRs as a biomarker for schizophrenia while paying detailed attention to the clinical and methodological factors that could contribute to variability in results across studies.

## 2. Materials and Methods

### 2.1. Literature Search Strategy

This meta-analysis followed the Reporting Items for Systematic Reviews and Meta-Analyses (PRISMA) guidelines [22] (Appendix A). Four databases were searched: MedLine, Scopus, Embase, and the Web of Science. To build upon the work of the last comprehensive meta-analysis by Thuné et al. [16], which ceased its literature search in March 2016, we conducted an updated search extending from March 2016 to June 2023. The combination of keywords included auditory (or gamma or 40 Hz) steady-state response and schizophrenia (or psychosis or psychoses or psychotic disorder or schizophrenic disorder).

### 2.2. Study Selection

The initial inclusion criteria were also based on the previous meta-analysis [16]: (1) articles written in French or English, (2) human studies, (3) original research articles with new data, (4) studied that used EEG or MEG to measure ASSRs, (5) studies with at least one sample of patients with schizophrenia or an at-risk population and one sample of healthy controls, (6) studies containing measures of spectral evoked power and/or intertrial phase coherence (ITPC, also known as phase-lock factor and phase synchronization), and (7) studies that included sufficient statistical information (sample sizes and mean values and/or raw data and/or *p*-values and/or effect sizes). Exclusion criteria were as follows: (1) published before March 2016; (2) full-text not available; and (3) duplicates. The selection of studies was based on a consensual decision by IZ and SP.

The collected articles were imported into the EndNote 20 software, and throughout the selection process, which followed the PRISMA model, the two investigators engaged in ongoing consultation to make their decisions.

### 2.3. Data Extraction

The two investigators decided on the charting form as follows: authors, publication year, sample size for each population, percentage of males within each population, age, illness duration (mean, sd), chlorpromazine dosage in mg/day (mean, sd), PANS scores or scores converted from SANS or SAPS to PANSS via an open-access website (http://converteasy.org, accessed on 15 August 2023), image analysis level (sensor and/or source), stimulus type (click or amplitude), duration of stimulus presentation (brief ≤ 500 ms or long ≥ 1000 ms), attention load (characterized as passive, active, or involving distraction), and analyzed frequencies (40, 30 or 20 Hz). The populations include Healthy Controls (HC), individuals with Schizophrenia (SC), individuals with First Episode Psychosis (FEP), and At-Risk populations (AR), which consisted of both clinical at-risk individuals and relatives of the patients.

### 2.4. Statistical Analysis

Comprehensive Meta-Analysis Version-2 was used to calculate effect size estimates of the differences in ASSRs between schizophrenia patients (and at-risk individuals) and healthy volunteers [23]. The effect size estimates were calculated using Cohen’s d [24]. The direction of the effect size was considered positive if the ASSRs were reduced in schizophrenia patients (and at-risk individuals) relative to healthy individuals. Effect size estimates were calculated based on oscillation frequencies (40 Hz, 30 Hz, and 20 Hz) and ASSR components (power vs. phase). Following the convention of Cohen [24], effect size estimates of 0.2, 0.5, and 0.8 were considered small, medium, and large, respectively.

Aggregating effect size estimates across studies is more legitimate when estimates are homogeneous. Heterogeneity among effect size estimates was assessed with the Q statistics [25], with the magnitude of heterogeneity being evaluated with the I^2^ index [26]. The level of significance was set at *p* < 0.05. Given the heterogeneity of the meta-analysis (see below), a random-effects model was used for study aggregation. Relative to fixed-effects models, random-effects models take between-study variability into account and allow for population-level inferences [27].

To determine the potential impact on the results of methodological parameters, we performed sub-analyses on analysis type (sensor vs. source), stimulus types (click vs. amplitude), stimuli duration (brief vs. long), and attention load (active, distraction, and passive). Given the absence of an effect for beta oscillations at 20 Hz and the limited studies focusing on gamma oscillations at 30 Hz, the sub-analyses were confined to studies that examined 40 Hz gamma oscillations. To estimate the effect of sociodemographic variables [e.g., age of participants (in years) and sex ratio (% of males)], antipsychotic dosage (e.g., mean chlorpromazine equivalents) and psychiatric (positive and negative) symptoms on results, we performed meta-regression analyses. Finally, the possibility of publication bias was examined with Egger’s regression test [28], Begg and Mazumdar’s rank correlation test [29], and a visual inspection of the funnel plot. As per guidelines from Cochrane (https://cochrane.org, accessed on 15 August 2023), we performed the sub-analyses, meta-regression, and publication bias analyses only for outcomes comprising at least 10 study arms.

Some studies reported negative findings without providing values (t-, *p*-, means and SD, etc.), allowing us to calculate effect size estimates. Based on an approach described in Appendix B, the effect size estimates for these studies were imputed. As per guidelines from Cochrane (https://cochrane.org), we aggregated results across studies with and without the imputed effect size estimates.

## 3. Results

### 3.1. Sample

A total of 37 publications were included in the meta-analysis (Appendix A). (*n* = 480) Based on our criteria, 70 articles were excluded (healthy controls only or patients only [*n* = 29], data re-analyzed in later studies [*n* = 9], full text not available [*n* = 28] and no power or phase results [*n* = 5]). In cases where datasets from the earlier studies had either been reused or developed in more recent studies, we retained the datasets with the largest sample sizes for a more robust analysis. This strategy was notably true for the dataset initially presented by Tada et al. [30], which was subsequently expanded by Koshiyama et al. [31], for nine recent studies that have since been updated with larger sample sizes.

The total sample consists of 1788 individuals diagnosed with schizophrenia, with 208 of them experiencing their first episode of psychosis,281 individuals identified as at-risk (either relatives or clinical high-risk individuals), along with 1603 healthy controls (Appendix A). Each included study provided data on ASSR power and/or phase, recorded at the sensor or source level. The stimuli used were either click or amplitude-modulated tones, and the studies varied in attention load and stimulus duration. Notably, results for 40 Hz oscillations were reported in all included studies (see Table 1).

### 3.2. Differences in ASSRs between Groups

#### 3.2.1. Gamma Oscillations at 40 Hz

For gamma oscillations, the ASSRs were reduced in schizophrenia patients relative to controls, and the magnitude of this reduction was in the moderate range (Table 2). This was the case for analyses based on both power (28 study arms) and phase (23 study arms) components of ASSRs. For both power and phase, there was evidence of publication bias, as determined by Egger’s test (power: t = 2.0; *p* = 0.057; phase: t = 3.1; *p* = 0.005; Appendix A) and the Begg and Mazumbar’s test (power: Kendall’s tau = 0.392; *p* = 0.003; phase: Kendall’s tau = 0.454; *p* = 0.002; Appendix A). With the addition of the imputed value of negative findings, the estimates remained moderate in both cases (power and phase) for comparing schizophrenia patients and controls (Table 2).

In individuals with a first episode of psychosis, moderate-to-large effects were observed for the power and phase components of the ASSRs (8 and 4 study arms), respectively (Table 2). In individuals at risk for psychosis, a non-significant and small effect was observed for the power component of the ASSRs (6 study arms), while a small-to-moderate reduction was observed in the case of the phase component of the ASSRs (4 study arms) (Table 2).

#### 3.2.2. Gamma Oscillations at 30 Hz

For gamma oscillations (30 Hz), small-to-moderate reductions in the ASSRs were observed in schizophrenia patients relative to controls for both power (7 study arms) and phase (5 study arms) (Table 2).

#### 3.2.3. Beta Oscillations at 20 Hz

For beta oscillations, no differences in the ASSRs were observed between schizophrenia patients and healthy volunteers, regardless of power and phase (Table 2). There was no evidence of publication bias (power; 11 study arms) using both Egger’s test (t = 1.6; *p* = 0.153) and Begg and Mazumbar’s test (Kendall’s tau = −0.127; *p* = 0.586) (Appendix A).

### 3.3. Sub-Analyses

Due to a lack of effect for beta oscillations at 20 Hz and a few studies examining gamma oscillations at 30 Hz, sub-analyses were only performed with studies examining 40 Hz gamma oscillations. Overall, for the power component of ASSRs, image analysis type, stimulus type, stimuli duration, and attention load had small effects on the results (Table 3). The reduction in ASSRs in schizophrenia patients was larger in studies using source analyses than analyses of sensors. The reduction in ASSRs was more prominent in studies using click trains instead of amplitude-modulated tones. The reduction in ASSRs was more significant in studies using brief rather than long stimuli. Finally, it is important to note that the reduction in ASSRs in schizophrenia patients was more pronounced in studies that used a distracting attention load. However, this difference was observed based on only 2 study arms.

As for the phase component of the ASSRs, there were no significant effects of analysis type, stimulus type, or the duration of stimuli on gamma oscillation results (Table 3). The reduction in ASSRs in schizophrenia patients was larger in studies that used a distracting attention load; however, this difference was observed based on only 2 study arms.

### 3.4. Meta-Regression Analyses

Due to a lack of effect for beta oscillations at 20 Hz and a small number of studies examining gamma oscillations at 30 Hz, sub-analyses were only performed with studies examining 40 Hz gamma oscillations. For the power component of ASSRs, there were no significant associations between age, sex ratio, positive symptoms, negative symptoms, chlorpromazine equivalents, or the magnitude of ASSR reduction observed in schizophrenia (Table 4).

For the phase component, a significant and negative association was observed between patients’ age and the magnitude of the ASSR reduction, such that the reduction in gamma oscillations was more robust in younger patients (Table 4; Figure 1). None of the other clinical variables were significantly associated with the magnitude of the reduction in ASSRs observed in schizophrenia (Table 4).

## 4. Discussion

Our updated meta-analysis aimed to provide an overview of ASSRs as potential biomarkers of schizophrenia by synthesizing previous data with recent ones, focusing on previously unexplored clinical populations, frequencies, and variables. Our main findings include moderate ASSR reductions in gamma oscillations at 40 Hz, a subtle effect at 30 Hz in schizophrenia patients, no discernible difference in beta oscillations at 20 Hz, and distinct associations based on attentional load and age.

### 4.1. ASSRs in Different Clinical Stages of Schizophrenia

In patients with schizophrenia, a significant effect in the gamma range at 40 Hz was observed, shedding light on the role of gamma-band abnormalities in the pathophysiology of schizophrenia, potentially indicating deficits in the integration of cortical activity. The modest effect at 30 Hz and not at 20 Hz suggests a possible frequency-specific nature of ASSR deficiencies in schizophrenia, which needs further exploration. A publication bias was found when examining auditory steady-state responses (ASSRs) in gamma oscillations’ power and phase components at 40 Hz.

In individuals at risk for psychosis, only a small effect was seen in the power component, while the phase component showed a small-to-moderate reduction. Conversely, in the context of patients experiencing their first episode of psychosis, the observed moderate-to-large effects on both the power and phase components of the 40 Hz ASSR suggest a significant reduction in auditory processing capabilities during these early stages of the disorder. This pattern contrasts with the anticipated gradient of ASSR alterations (SCZ > FEP > at risk), typically seen in other potential biomarkers of schizophrenia, such as neurobiological and neurocognitive markers [18].

Gamma oscillations are known to be dependent on glutamatergic mechanisms [3,4]. Interestingly, a meta-analysis from Nakahara et al. [67] revealed elevated levels of glutamate in limbic and subcortical brain regions that were particularly prominent in individuals with early stage schizophrenia. In theory, it can be hypothesized that deficits in gamma oscillations are related to the pronounced glutamatergic alterations observed in the early stages of psychosis [68,69]. However, this hypothesis would need to be tested. Alternatively, our results could be explained by the effect of antipsychotics. Indeed, the pronounced ASSR deficits observed in FEP patients could be due to a relative normalization of gamma oscillations with long-term exposure to antipsychotics in chronic patients. This remains speculative, however, as the literature on the effect of antipsychotics on gamma oscillations is still in its infancy [49].

### 4.2. Sub-Analyses

Regarding 40 Hz-gamma oscillations, variations in analysis types, stimulus types, and durations were observed, with more significant reductions in ASSR in certain conditions, notably source analyses, click trains, and studies based on distraction. The 40 Hz ASSR power decrease in schizophrenia patients was more pronounced in studies using source analyses than in sensor analyses. Sensor-based analyses, mainly using EEG electrodes at Cz and Fz, give a broader view of neuronal activity, reflecting signals dispersed over the skull. While effective in registering ASSRs due to their fronto-central location, these sensors might miss subtleties from specific regions like the auditory cortex. Importantly, ASSR sources are not limited to the auditory cortex but also encompass areas like the cerebellum, frontal, and parietal lobes [70,71]. Conversely, source-based analyses target auditory regions, offering a detailed and localized perspective. Edgar et al. [19] point out that this approach can discern complex hemispheric response variations at 40 Hz, offering more region-specific insights. Therefore, the pronounced reduction observed in ASSR power from source-level analyses can be attributed to its precision in targeting ASSR origins, whereas sensor-level analysis, capturing a broader, averaged view, may not reflect the reduction to the same extent. It must be noted, however, that the difference between the source and sensor was small (power: d = 0.663 vs. d = 0.483; phase: d = 0.746 vs. d = 0.553), but it was observed in the case of both the power and phase components of the ASSR.

Studies employing click trains over amplitude-modulated tones saw a more extensive 40 Hz ASSR reduction in the case of the power component. According to Griskova-Bulanova et al. [52], click stimuli generated more powerful and better-coordinated ASSRs. The underlying reason could be the nature of the stimuli: clicks, characterized by a brief broadband noise with rapid transitions, engage larger regions of the auditory cortex. This broad engagement, combined with the abruptness of clicks, results in more robust ASSR responses. By contrast, amplitude-modulated tones have slower transitions and target a narrower frequency range, which could lead to milder ASSR results [72].

Regarding the influence of stimulus duration on 40 Hz ASSR effects (power and phase components), our findings indicate a trend where the reduction in ASSRs was more pronounced in studies utilizing brief stimuli compared to those with longer durations. Hamm et al. [20] reported that decreases in 40 Hz training in schizophrenia were more noticeable with 500 ms stimulus trains than with 1500 ms trains. This further underlines the possible influence of stimulus duration on the effects of ASSRs, though the overall difference was relatively small (d ≈ 0.15).

A significant observation was made regarding attention load. Studies focusing on distraction showed a more pronounced reduction in ASSRs in power and phase components. While this effect appears to be primarily attributed to distraction, it is essential to note that this observation is derived from only two studies. Given this limited dataset, we should interpret these findings with caution. This underscores the need for additional research to validate these results.

### 4.3. Meta-Regression Analysis

Our meta-regression analysis did not reveal significant associations between ASSR power reduction in schizophrenia patients and variables such as age, sex ratio, types of symptoms, or medication dosages. Interestingly, our data indicate that younger schizophrenia patients exhibit a more pronounced 40 Hz ASSR phase. This finding seems at odds with research suggesting an age-related neurobiological deterioration in schizophrenia [73]. Yet, it aligns with our observation of a greater ASSR reduction in First Episode Psychosis (FEP) patients than in chronic cases. Such results underscore the need for more extensive research across various age groups and stages of schizophrenia. As such, our results extend those from the previous meta-analysis by Thuné et al. [16], which also pinpointed the significant impact of age, with younger patients (≤39.8 years) showing a trend toward more robust ASSR reductions compared to older ones (>39.8 years). Given that only three studies drove the association between age and the 40 Hz ASSR phase, there is a need for more research across varied age groups and stages of schizophrenia.

### 4.4. Strengths and Limitations

Building on prior work by Thuné et al. [16], our meta-analysis significantly expands the scope of the research by incorporating a more extensive and diverse set of clinical populations, including recent data on first-episode psychosis and at-risk populations. It offers a comprehensive update, examining stimulus duration, type, and level of analysis, and uniquely addresses under-explored factors such as symptom severity, medication dosage, and attentional focus, advancing our understanding of ASSRs in schizophrenia. Despite these strengths, the current meta-analysis has a few limitations that need to be acknowledged. First, the insufficiency of studies, particularly in 30 Hz and at-risk populations, is a limitation that prevents the validation of the observed trends. Furthermore, the paucity of research on different methodologies limited our ability to investigate the effects of right vs. left hemisphere involvement and distinctions between time response segments when comparing ASSRs across groups. In addition, publication bias suggests a predisposition toward publishing studies with significant effects, which could distort the results. We incorporated the non-significant results into our models using our custom imputation technique to address this issue. Even with the inclusion of these imputed negative results, our overall results did not change. Finally, the lack of measurement of specific positive symptoms, such as auditory hallucinations, may have masked potential associations between ASSRs and symptoms. Current bottom-up models have proposed that abnormalities in the early processing of auditory stimuli could be involved in the emergence of auditory hallucinations [2]. Only one study has examined ASSRs in schizophrenia patients who experience auditory hallucinations and those who do not [48], but it unexpectedly failed to show differences between these two subgroups.

## 5. Conclusions

Our meta-analysis, encompassing recent and diverse data, reinforces the potential of auditory steady-state responses (ASSRs) as biomarkers of schizophrenia. Key findings include moderate reductions in gamma oscillations at 40 Hz and subtle effects at 30 Hz, with no significant changes at 20 Hz. In particular, patients suffering from a first episode of psychosis showed pronounced reductions in ASSRs, contrasting with the less severe changes observed in at-risk individuals. Furthermore, variations in ASSRs related to analysis type, stimulus, duration, and attention load highlight the complex influence of methodological factors on capturing auditory processing abnormalities in schizophrenia. Future research directions should focus on expanding the scope of the study to incorporate and differentiate clinical populations, such as at-risk individuals and those with first-episode psychosis. Also, more studies at frequencies other than 40 HZ could help confirm the specificity of ASSR impairment at 40 Hz in schizophrenia. There is also a need for a detailed exploration of specific positive symptoms, particularly auditory hallucinations. In addition, future studies could focus more on exploring conditions of attention and distraction, given their potential notable impact on ASSRs, as suggested in this research. As more pronounced effects were observed with source analysis, click and brief stimuli, future research should consider using these methods to obtain more conclusive and significant results.

## Figures and Tables

**Figure 1 brainsci-13-01722-f001:**
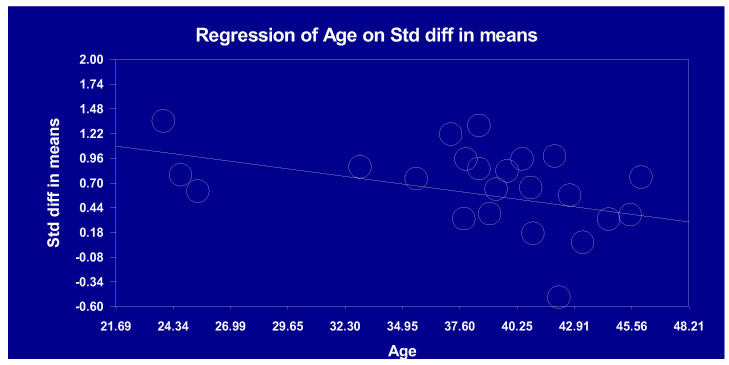
Linear association between age and auditory steady-state response (40 Hz; phase).

**Table 1 brainsci-13-01722-t001:** Methodological characteristics of the selected studies.

Authors	Image Analysis Level	Stimulus Type	Stimulus Duration *	Attention Load	Analyzed Frequencies (in Hz)	Power Results	Phase Results
Kwon et al., 1999 [32]	Sensor	Click	Brief	Passive	20, 30, 40	Yes	Yes
Hong et al., 2004 [33]	Sensor	Click	Brief	Passive	20, 30, 40	Yes	
Spencer et al., 2008 [34]	Sensor	Click	Brief	Passive	20, 30, 40	Yes	Yes
Vierling-claasen et al., 2008 [35]	Source	Click	Brief		20, 30, 40	Yes	
Teale et al., 2008 [36]	Source	Amplitude	Brief	Distraction	40		Yes
Wilson et al., 2008 [37]	Source	Click	Brief	Distraction	20, 30, 40	Yes	
Spencer et al., 2009 [38]	Source and Sensor	Click	Brief	Passive	40	Yes	Yes
Hamm et al., 2011 [39]	Source	Amplitude	Long	Active	5, 20, 40, 80, 160		Yes
Tsuchimoto et al., 2011 [40]	Sensor	Click	Brief	Passive	20, 30, 40, 80	Yes	Yes
Hamm et al., 2012 [41]	Sensor	Amplitude	Long	Passive	16 to 44	Yes	Yes
Kömek et al., 2012 [42]	Sensor	Click	Brief	Active	20, 30, 40	Yes	
Rass et al., 2012 [43]	Sensor	Click	Brief	Passive	20, 30, 40, 50	Yes	Yes
Kirihara et al., 2012 [44]	Sensor	Click	Brief	Passive	20, 30, 40		Yes
Roach et al., 2013 [45]	Sensor	Click	Brief	Passive	20, 30, 40		Yes
Edgar et al., 2014 [46]	Source	Amplitude	Long		4–12, 40		Yes
Hirano et al., 2015 [47]	Source	Click	Brief	Passive	20, 30, 40	Yes	Yes
Hamm et al., 2015 [20]	Sensor	Amplitude	Brief and long	Active and passive	40	Yes	
Griskova-Bulanova, 2016 [48]	Sensor	Click	Brief	Passive	40		Yes
Alegre, 2017 [49]	Sensor	Chirps	Long	Passive	40, 100	Yes	Yes
Light, 2017 [50]	Sensor	Click	Brief	Passive	40	Yes	Yes
Edgar, 2018 [51]	Source and sensor	Amplitude	Long		4–16, 40	Yes	Yes
Griskova-Bulanova, 2018 [52]	Sensor	Click and amplitude	Brief	Distraction	40	Yes	Yes
Koshiyama, 2018 [31]	Sensor	Click	Brief	Passive	40	Yes	Yes
Puvvada, 2018 [53]	Sensor	Click		Passive	2.5, 5, 10, 20, 40, 80	Yes	Yes
Sun, 2018 [54]	Sensor	Click	Brief	Passive	20, 40	Yes	Yes
Wang, 2018 [55]	Sensor	Click	Brief	Passive	40	Yes	Yes
Zhou, 2018 [56]	Sensor	Click	Brief	Passive	20, 30, 40	Yes	Yes
Bartolomeo, 2019 [57]	Sensor	Click	Brief	Passive	40	Yes	
Kim, 2019 [58]	Source and sensor	Click	Brief	Passive	40	Yes	Yes
Parker, 2019 [59]	Sensor	Amplitude	Long	Active	20, 40, 80	Yes	Yes
Lepock, 2020 [60]	Sensor	Click	Brief	Passive	40	Yes	Yes
Murphy, 2020 [61]	Source	Click	Long	Active	20, 30, 40	Yes	Yes
Grent-’t-Jong, 2021 [62]	Source	Amplitude	Long	Active	40	Yes	Yes
Koshiyama, 2021 [63]	Sensor	Click	Brief		40	Yes	Yes
Coffman, 2022 [64]	Sensor	Click	Brief	Active and Distraction	40	Yes	Yes
Du, 2023 [65]	Source	Click	Brief	Passive	40	Yes	
Ogyu, 2023 [66]	Sensor	Click	600 ms	Passive	40	Yes	Yes

* Stimulus duration (brief ≤ 500 ms or long ≥ 1000 ms).

**Table 2 brainsci-13-01722-t002:** Main results.

		Study Arms	Cohen’s d	*p*-Value	95% Confidence Interval	Q-Test	I^2^
40 Hz							
Schizophrenia	Power	28	0.538	0.0001	(0.391 to 0.684)	Q = 66.6; *p* = 0.0001	59.5%
Schizophrenia	Phase	23	0.581	0.0001	(0.432 to 0.730)	Q = 55.7; *p* = 0.0001	60.5%
First episode	Power	8	0.705	0.0001	(0.387 to 1.023)	Q = 14.6; *p* = 0.041	52.2%
First episode	Phase	4	0.919	0.0001	(0.584 to 1.253)	Q = 3.7; *p* = 0.300	18.2%
At risk	Power	6	0.243	0.123	(−0.066 to 0.552)	Q = 13.9; *p* = 0.016	64.0%
At risk	Phase	4	0.360	0.003	(0.124 to 0.596)	Q = 4.0; *p* = 0.262	24.9%
40 Hz with imputed non-significant results							
Schizophrenia	Power	30	0.515	0.0001	(0.378 to 0.652)	Q = 67.8; *p* = 0.0001	57.2%
Schizophrenia	Phase	30	0.503	0.0001	(0.383 to 0.504)	Q = 58.6; *p* = 0.001	50.5%
30 Hz							
Schizophrenia	Power	7	0.295	0.018	(0.051 to 0.538)	Q = 6.8; *p* = 0.341	11.6%
Schizophrenia	Phase	5	0.372	0.001	(0.145 to 0.598)	Q = 1.5; *p* = 0.833	0%
20 Hz							
Schizophrenia	Power	11	0.085	0.380	(−0.104 to 0.274)	Q = 16.1; *p* = 0.098	37.8%
Schizophrenia	Phase	9	0.083	0.221	(−0.050 to 0.216)	Q = 6.7; *p* = 0.568	0%

**Table 3 brainsci-13-01722-t003:** Sub-analyses of gamma oscillations (40 Hz) for both the power and phase components of the auditory steady-state response.

Variable		Study Arms	Cohen’s d	*p*-Value	95% Confidence Interval	Q-Test	I^2^
Power							
Analysis	Sensor	21	0.488	0.0001	(0.319 to 0.658)	Q = 55.8; *p* = 0.0001	64.2%
	Source	6	0.712	0.0001	(0.393 to 1.031)	Q = 8.1; *p* = 0.151	38.2%
Stimuli	Amplitude	4	0.319	0.202	(−0.172 to 0.810)	Q = 12.3; *p* = 0.006	75.7%
	Click	22	0.550	0.0001	(0.388 to 0.713)	Q = 49.8; *p* = 0.0001	57.9%
Duration	Brief	20	0.542	0.0001	(0.340 to 0.744)	Q = 59.1; *p* = 0.0001	67.8%
	Long	6	0.394	0.030	(0.037 to 0.752)	Q = 12.0; *p* = 0.034	58.4%
Attention	Active	5	0.483	0.0001	(0.285 to 0.680)	Q = 3.2; *p* = 0.522	0%
	Distraction	2	1.097	0.0001	(0.576 to 1.618)	Q = 0.5; *p* = 0.492	0%
	Passive	18	0.451	0.0001	(0.248 to 0.655)	Q = 48.1; *p* = 0.0001	64.6%
Phase							
Analysis	Sensor	19	0.553	0.0001	(0.389 to 0.716)	Q = 51.1; *p* = 0.0001	64.8%
	Source	3	0.746	0.0001	(0.409 to 1.083)	Q = 0.4; *p* = 0.808	0%
Stimuli	Amplitude	3	0.518	0.0001	(0.235 to 0.800)	Q = 2.6; *p* = 0.271	23.4%
	Click	18	0.564	0.0001	(0.389 to 0.740)	Q = 49.3; *p* = 0.0001	65.5%
Duration	Brief	18	0.655	0.0001	(0.463 to 0.847)	Q = 51.0; *p* = 0.0001	66.7%
	Long	3	0.459	0.0001	(0.243 to 0.676)	Q = 1.8; *p* = 0.398	0%
Attention	Active	1	0.375	0.003	(0.124 to 0.626)	---	---
	Distraction	2	0.969	0.0001	(0.491 to 1.446)	Q = 0.004; *p* = 0.949	0%
	Passive	18	0.606	0.0001	(0.407 to 0.805)	Q = 48.5; *p* = 0.0001	65.0%

**Table 4 brainsci-13-01722-t004:** Meta-regression analyses for studies assessing gamma oscillations (40 Hz).

Predictor	Study Arms	Slope	95% CI	*p*-Value
Power component of ASSRs				
Age	28	−0.016	(−0.035 to 0.004)	0.114
Sex ratio	27	0.005	(−0.003 to 0.012)	0.200
Positive symptoms	18	0.010	(−0.055 to 0.076)	0.760
Negative symptoms	18	0.019	(−0.026 to 0.065)	0.409
Chlorpromazine equivalents	19	0.0004	(−0.0008 to 0.002)	0.485
Phase component of ASSRs				
Age	23	−0.031	(−0.052 to −0.008)	0.008
Sex ratio	22	−0.023	(−0.048 to 0.003)	0.082
Positive symptoms	18	0.049	(−0.011 to 0.110)	0.108
Negative symptoms	18	0.036	(−0.015 to 0.086)	0.170
Chlorpromazine equivalents	18	−0.0003	(−0.002 to 0.001)	0.635

ASSRs: Auditory steady-state responses; CI: Confidence interval.

## Data Availability

No new data were created or analyzed in this study. Data sharing is not applicable to this article.

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
