# Peer review of "Auditory Steady-State Responses in Schizophrenia: An Updated Meta-Analysis"

_brainsci, 2023, doi:10.3390/brainsci13121722_

Round 1

Reviewer 1 Report

Comments and Suggestions for Authors

This article is well-organized for an updated meta-analysis of previous reports on ASSR in patients with schizophrenia. However, the following details should be considered for revision.

Page 7, lines 219.

Please revise the following sentence for clarification. “Finally, the decrease in ASSR was more important in studies based on distraction.”

Page 8, line 241.

There is an error in the table title. It should be 'Table 4,' not 'Table 5’.

Page 9, lines 268-278.

This part of the discussion appears disconnected from the analysis conducted in this article. Although the authors discuss glutamine and glutamate levels in unmedicated and medicated patients with schizophrenia, it seems that this paper does not analyze changes in gamma oscillations in unmedicated patients.

Author Response

This article is well-organized for an updated meta-analysis of previous reports on ASSR in patients with schizophrenia. However, the following details should be considered for revision.

Thank you for your valuable feedback. We appreciate your constructive comments and will carefully consider your suggested revisions to further improve the quality of our manuscript.

Page 7, lines 219.

Please revise the following sentence for clarification. “Finally, the decrease in ASSR was more important in studies based on distraction.”

            We have revised the sentence for greater clarity.

Page 8, line 241.

There is an error in the table title. It should be 'Table 4,' not 'Table 5’.

            Thank you for pointing out this error, it has been corrected.

Page 9, lines 268-278.

This part of the discussion appears disconnected from the analysis conducted in this article. Although the authors discuss glutamine and glutamate levels in unmedicated and medicated patients with schizophrenia, it seems that this paper does not analyze changes in gamma oscillations in unmedicated patients.

Thank you for your insightful comment. We have revised the discussion to align with our study’s findings more closely. We now emphasize the theoretical link between gamma oscillations and glutamatergic activity, particularly in early-stage schizophrenia, based on Nakahara et al.’s meta-analysis. We also acknowledge the potential influence of antipsychotics on these oscillations, noting the need for further research in this area. This revision aims to address your concern regarding the direct relevance of our discussion to the analyzed data.

Reviewer 2 Report

Comments and Suggestions for Authors

The authors presented an interesting overview of Auditory Steady-State Response in Schizophrenia. Overall, the paper is well-executed, and I have only minor feedback.

I'd like to propose enhancing the quality and refining the caption content of Figure 1,2,3 and 4.

The labels within Table 1 in the supplementary material require improvement.

I would recommend a few minor grammar and language corrections. Additionally, there are some typos that need to be addressed.

Author Response

The authors presented an interesting overview of Auditory Steady-State Response in Schizophrenia. Overall, the paper is well-executed, and I have only minor feedback.

Thank you for your comments. We are grateful for your positive feedback.

I'd like to propose enhancing the quality and refining the caption content of Figure 1,2,3 and 4.

As per your recommendation, we have refined the captions to include more precise descriptions of the axes and the scientific content depicted in the funnel plots for Figures 1, 2, 3, and 4.

The labels within Table 1 in the supplementary material require improvement.

We have taken steps to minimize the use of abbreviations for clarity and ease of understanding. This should enhance the accessibility and readability of the data presented in the table.

I would recommend a few minor grammar and language corrections. Additionally, there are some typos that need to be addressed.

Thank you for your feedback. We have revised and corrected several sentences for clarity and consistency. Additionally, we have standardized the use of 'ASSR' and 'ASSRs'.